# Towards Biological Continual Learning with Spiking Hopfield Networks

## Abstract

Modern Hopfield networks are often viewed as biologically inspired associative memories, yet they lack the spiking dynamics and local learning rules that underpin real neural computation. In this work, we introduce a Spiking Hopfield Network (SHN) that incorporates discrete spike-based communication and a spike-timing–dependent plasticity (STDP) rule, enhancing biological plausibility while retaining the network's capacity for online learning. To further support continual updates, we propose an Elastic Weight Consolidation (EWC)–inspired mechanism adapted to this local learning setting, reducing catastrophic forgetting. Together, these contributions yield a lightweight and biologically grounded framework that combines efficient memory retrieval with resilience to continual adaptation.

## 1 Introduction

Deep neural networks (DNNs) trained with backpropagation dominate modern AI, powering transformers, large language models, and diffusion models. While these systems deliver unmatched performance on large-scale tasks, their training is computationally expensive, centralized, and fundamentally offline: once deployed, models remain fixed until retrained in future batches. This paradigm leaves a complementary need largely unaddressed—lightweight models that can adapt continuously to new data without relying on repeated global retraining (Hoffpauir et al., 2023).

Seemingly in contrast, the human brain achieves real-time learning and recall with remarkable energy efficiency. A central structure in this process is the hippocampus, which supports episodic memory by reconstructing experiences when triggered by partial cues (Eichenbaum, 2017; Moscovitch et al., 2016; Casanueva-Morato et al., 2024). Modern Hopfield Networks (MHNs) (Krotov & Hopfield, 2016; Ramsauer et al., 2020) capture aspects of this ability by retrieving stored patterns from incomplete input, making them a valuable step toward biologically inspired memory models.

Yet MHNs remain far from biological realism. Neurons communicate through discrete spikes, and their adaptation is governed by local Hebbian plasticity such as spike-timing–dependent plasticity (STDP) (Markram et al., 2012)—mechanisms missing from existing formulations. As a result, MHNs capture structural analogies to hippocampal memory but omit the essential ingredients of spiking dynamics and local learning that support fast retrieval and continual adaptation. To further complicate online use, the challenge of catastrophic forgetting (French, 1999) remains unresolved: current MHNs, and even STDP alone, cannot preserve old patterns while learning new ones. Prior approaches such as Elastic Weight Consolidation (EWC) (Kirkpatrick et al., 2017) mitigate forgetting in gradient-based models but are incompatible with STDP, where backpropagation is absent. These gaps motivate our framework, which integrates spiking dynamics and local STDP with an EWC-inspired mechanism tailored to continual associative memory.

In this paper, we present a biologically inspired variation of the MHN, called the Spiking Hopfield Network (SHN), designed to support lightweight online learning. Our main contributions are:

1. **Spiking representation.** We integrate spiking dynamics into the MHN framework, enabling memory storage and recall through discrete spike events in a biologically plausible manner.

2. **Retrieval algorithm.** We develop a fully spike-based recall mechanism that performs competitively with the standard Hopfield retrieval rule, providing an effective and biologically consistent approach to memory reconstruction.

3. **Forgetting mitigation.** We develop an Elastic Weight Consolidation (EWC)–inspired method adapted to local STDP, which mitigates catastrophic forgetting during continual updates while remaining entirely gradient-free.

Together, these contributions offer a complementary path to mainstream backpropagation-driven AI: a lightweight, biologically grounded framework for online learning that combines efficient retrieval with resilience to continual adaptation.

## 2 RELATED WORKS AND PRELIMINARIES

To design our biologically inspired Spiking Hopfield Network for online learning, we turn away from traditional gradient-based techniques and instead draw on theories that resemble biological processes memeory structure in the human brain. The following background elements serve as the building blocks and inspiration for our approach.

### 2.1 ADAPTIVE LEAKY INTEGRATE-AND-FIRE (ALIF) NEURON

A fundamental component in our work is the adaptive leaky integrate-and-fire (ALIF) neuron (Bellec et al., 2018; Dayan et al., 2003), which models membrane potential $u$ that decays over a timescale $\tau_m$ but is continually driven by synaptic input $I(t)$ through resistance $R$:

$$\tau_m \frac{du}{dt} = -u + RI(t).$$

(1)

Unlike a fixed threshold, the firing threshold $\theta(t)$ adapts to recent activity: after each spike at time $t_s$, it is incremented by $\beta$, and each increment then decays exponentially with time constant $\tau_\theta$, so that in the absence of further spikes the threshold approaches the baseline level $\theta_0$:

$$\theta(t) = \theta_0 + \beta \sum_{t_s < t} \exp\left(-\frac{t-t_s}{\tau_\theta}\right).$$

(2)

This coupling of membrane decay and adaptive threshold captures refractoriness while retaining computational efficiency.

### 2.2 LOCAL LEARNING THROUGH SPIKE-TIME–DEPENDENT PLASTICITY

A biologically grounded alternative to gradient-based optimization is spike-timing–dependent plasticity (STDP), where synaptic updates depend only on the relative timing of pre- and postsynaptic spikes (Bi & Poo, 1999). The weight change is defined as:

$$\Delta w_{ij} = \begin{cases} A_+ \exp(-\frac{\Delta t}{\tau_+}), & \Delta t > 0, \\ -A_- \exp(\frac{\Delta t}{\tau_-}), & \Delta t < 0, \end{cases}$$

(3)

with $\Delta t = t_{\text{post}} - t_{\text{pre}}$, and parameters $A_\pm, \tau_\pm$ setting the update scale and time constants. Presynaptic spikes that precede postsynaptic firing lead to Long-Term Potentiation (LTP), while the reverse ordering induces Long-Term Depression (LTD).

This temporally asymmetric rule enables local adaptation of synaptic weights, allowing Hopfield memories to be updated in an online fashion.

### 2.3 MODERN HOPFIELD NETWORKS (MHN) FOR EFFICIENT STORAGE AND RECALL

Similar to the role of the hippocampus in managing episodic memory, MHNs retrieve stored patterns through an energy formulation (Demircigil et al., 2017):

$$E = -\text{lse}(\beta, X^t\xi) + \tfrac{1}{2}\xi^T\xi + \beta^{-1}\log N + \tfrac{1}{2}M^2.$$

(4)

which can be reformulated via the Concave–Convex Procedure (CCCP) (Yuille & Rangarajan, 2001) into the retrieval update rule (Ramsauer et al., 2020):

$$\xi_{\text{new}} = X \,\text{softmax}(\beta X^\top \xi).$$

(5)

In this setting, the weight matrix $X^\top$ represents memory storage (Hopfield, 1982; Krotov & Hopfield, 2020), as shown in Figure 1(a), while the scaling factor $\beta$ modulates recall selectivity across the energy landscape. Although this formulation is efficient and guarantees convergence, prior work has shown that dense storage can produce spurious attractors in the energy landscape (Figure 1(b)), leading retrieval toward unintended states (Hopfield, 1982; Krotov & Hopfield, 2020). We do not attempt to address this effect here; we note it only to highlight the broader challenge of ensuring stability in memory recall.

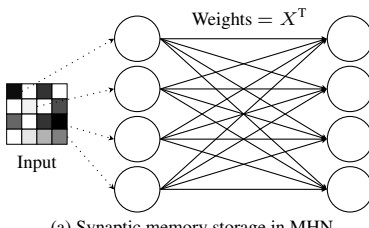

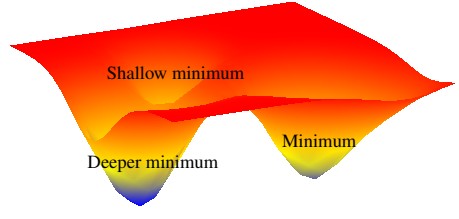

(a) Synaptic memory storage in MHN

(b) A high-density storage region with several minima

**Figure 1:** Illustrations of memory storage and retrieval in MHNs. (a) Input patterns are stored in the synaptic weights $X^\top$, where each output neuron maintains a weight representing a single input. (b) A region with high storage density can create spurious attractors, to which input data may converge, posing a constraint for the Hopfield recall algorithm.

## 2.4 ELASTIC WEIGHT CONSOLIDATION (EWC) AND CATASTROPHIC FORGETTING

Catastrophic forgetting arises when new inputs overwrite previously stored information during continual learning (Chen & Liu, 2022). Elastic Weight Consolidation (EWC) addresses this in gradient-trained networks by constraining important parameters to remain close to their past values (Kirkpatrick et al., 2017; Huszár, 2018), with the augmented loss

$$\mathcal{L}_{\text{EWC}} = \mathcal{L}_t + \tfrac{\lambda}{2} \sum_i F_i (\theta_i - \theta_i^*)^2, \tag{6}$$

where $\theta_i^*$ are parameters from past tasks, $F_i$ are diagonal Fisher information estimates, and $\lambda$ controls consolidation strength.

While MHNs can in principle be trained with gradient optimization, our biologically motivated setting employs local STDP updates, where synaptic weights serve directly as memory slots. In this context, global parameter regularization as in EWC could gradually disrupt stored memories and is not a natural fit for our design. Nonetheless, its central principle—preserving past knowledge while retaining adaptability—remains an important inspiration for adapting EWC-like constraints to slot-based memory updates in a Hopfield–STDP framework.

## 3 METHODOLOGIES

Our SHN extends MHN with spiking-based memory population and online retrieval. Its components and computational flow are introduced in the following subsections and illustrated in Figure 2 (a).

### 3.1 A SIMPLIFIED STDP (SIM-STDP) UPDATE RULE

Classical STDP Equation 3 accumulates weight changes over all spike pairs in a temporal window, while nearest-neighbor STDP (Song et al., 2000; Morrison et al., 2007) reduces this to the most recent presynaptic spike before a postsynaptic event. For online Hopfield updates, we streamline this further into a simplifed STDP (SIM-STDP):

$$\mathcal{W}(t_{\text{post}}) = \{\, t_p \mid t_{\text{post}} - T_{\text{win}} \leq t_p < t_{\text{post}} \,\}, \quad t^* = \max \mathcal{W}(t_{\text{post}}).$$

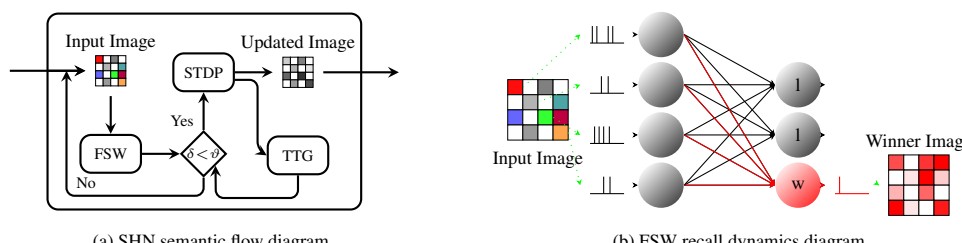

(a) SHN semantic flow diagram          (b) FSW recall dynamics diagram

**Figure 2:** I (a) SHN semantic flow diagram: an input image passes through FSW, which selects the winning neuron based on the threshold. The winner is updated via STDP, and TTG then renews its threshold using the winner's information. (b) FSW recall dynamics: for each new image, FSW selects the first neuron to fire as the winner (W). The winner's stored image (its weight vector) is updated via STDP, while losing neurons (L) are ignored, and FSW then proceeds to the next image.

where $\mathcal{W}(t_{\text{post}})$ denotes the set of presynaptic spikes falling within a window of length $T_{\text{win}}$ before the postsynaptic event. This leads to the following SIM-STDP update rule:

$$\Delta w_{ij}^{\text{LTP}} = \begin{cases} A_+ \exp(-\frac{t_{\text{post}}-t^*}{\tau_+}), & \mathcal{W}(t_{\text{post}}) \neq \varnothing, \\ 0, & \text{otherwise}, \end{cases} \tag{7}$$

$$\Delta w_{ij}^{\text{LTD}} = \begin{cases} -A_- \exp(-\frac{1}{\tau_-}), & \mathcal{W}(t_{\text{post}}) = \varnothing, \\ 0, & \text{otherwise}. \end{cases} \tag{8}$$

Thus, potentiation occurs only once, from the most recent presynaptic spike, while depression is applied as a fixed decrement if no spike exists in the window. This design preserves locality while avoiding unnecessary accumulation, yielding a lightweight, biologically inspired update suited for rapid online memory storage.

### 3.2 WTA-BASED MEMORY SLOT SELECTION

In our formulation, Hopfield storage is realized as neuron-specific slots: each output neuron with its afferent synapses serves as the substrate for one pattern. Updating all neurons indiscriminately would corrupt multiple slots at once. To avoid this, we couple the adaptive firing model with a Winner-Take-All (WTA) rule that assigns each input to a single dominant neuron.

At time $t$, the winner is

$$i^*(t) = \arg\max_i \{u_i(t) \mid u_i(t) > \theta_i(t)\}, \tag{9}$$

Here, $u_i(t)$ denotes the membrane potential and $\theta_i(t)$ the adaptive threshold from Equation 2. If no neuron exceeds the threshold, no update is performed, as illustrated in Figure 2(b

Only the synapses of $i^*(t)$ are then updated by SIM-STDP (Section 3.1), ensuring that storage remains sparse and slot-specific: the neuron that wins most consistently across timesteps becomes the stable representative of the input pattern. WTA thus provides the gating needed to preserve discrete memory assignments in our SHN.

### 3.3 FIRST SPIKE WINS (FSW): RETRIEVAL RULE

While storage in our SHN relies on WTA–STDP updates across timesteps, retrieval can be simplified. Instead of simulating full spike trains or tuning the $\beta$–softmax rule of MHNs Equation 5, we introduce the First Spike Wins (FSW) rule: the neuron that fires first is declared the winner, and recall terminates immediately as illustrated in Figure 2 (b).

Formally, let $t_i^{(1)}$ be the first spike time of neuron $i$, defined by $u_i(t_i^{(1)}) \geq \theta_i(t_i^{(1)})$. The FSW winner is

$$i^\dagger = \arg\min_i t_i^{(1)}. \tag{10}$$

**Remark 1.** *In our SHN pipeline, FSW provides the retrieval stage, complementing WTA–STDP storage. By avoiding reliance on $\beta$–softmax, it eliminates the need for parameter tuning and*

*yields sharper recall. Beyond our framework, FSW can serve as a lightweight retrieval rule in any Hopfield-style network, since it depends only on stored weights and firing thresholds.*

### 3.4 TEMPORAL THRESHOLD GATING (TTG) FOR CATASTROPHIC FORGETTING

In our SHN, synaptic weights are the memory substrate, so unconstrained STDP updates risk catastrophic forgetting: even small changes blur stored patterns. To prevent this, we introduce a Temporal Threshold Gating (TTG) mechanism inspired by—but distinct from—EWC. Instead of penalizing parameters globally, updates are gated locally by an adaptive threshold that decays over repeated activations of the same neuron.

Given an input $x$ and its winner neuron $i^*$ (identified by FSW), we compute the dissimilarity

$$\Delta = \mathcal{D}(x, W_{i^*}) \tag{11}$$

where $\mathcal{D}$ is a similarity measure (e.g. mean squared error). The update rule is

$$W_{i^*} \;\leftarrow\; \begin{cases} W_{i^*}^{\text{STDP}}, & \Delta \leq \vartheta_{i^*}, \\ W_{i^*}, & \text{otherwise}, \end{cases} \tag{12}$$

with $\vartheta_{i^*}$ the adaptive threshold. After each update, this threshold decays multiplicatively as

$$\vartheta_i \;\leftarrow\; \vartheta_i \Big( 1 - \tfrac{1}{\gamma + g(i)^n} \Big) \tag{13}$$

where $g(i)$ counts how many times neuron $i$ has previously won, $\gamma$ is a decay constant, and $n$ is an exponent that controls decay sharpness. While the formulation allows arbitrary $n$, we set $n = 1$ to ensure that the threshold $\vartheta$ decays smoothly without collapsing too quickly.

This design ensures that recently stored patterns remain plastic, while older ones become increasingly stable. In effect, only highly similar inputs can refresh a consolidated memory, preventing gradual blurring while still allowing selective adaptation.

The complete algorithmic steps are summarized in Algorithm 1, which shows how WTA, STDP, and decaying thresholds interact to mitigate catastrophic forgetting.

---

**Algorithm 1:** Threshold-based gating (TTG) update rule for SHN via STDP

---

**Input:** Input sample $x$
**Output:** Updated synaptic weights $W$, thresholds $\vartheta$
$i^* \leftarrow \text{WinnerNeuron}(x)$ ;                    /* winner via FSW Equation 10 */
$\Delta \leftarrow \mathcal{D}(x, W_{i^*})$ ;                    /* dissimilarity measure Equation 11 */
**if** $\Delta \leq \vartheta_{i^*}$ **then**
  $\quad W_{i^*} \leftarrow \text{STDP\_Update}(W_{i^*}, x)$;
**else**
  $\quad W_{i^*} \leftarrow W_{i^*}$ ;                    /* no change Equation 12 */
$\vartheta_{i^*} \leftarrow \vartheta_{i^*} \cdot (1 - \tfrac{1}{\gamma + g(i^*)^n})$ ;                    /* threshold update Equation 13 */

---

**Remark 2.** *Unlike classical EWC, which blends new and old information, TTG updates completely: patterns are either refreshed in full or left untouched. This preserves sharp, high-fidelity memories in SHN storage and avoids blur that would otherwise compromise retrieval under Hopfield recall or our FSW rule. It thus provides a natural complement to WTA slot selection, maintaining discrete and stable memories.*

## 4 EXPERIMENTS

### 4.1 EXPERIMENTAL SETUP

We evaluate our SHN on three datasets of increasing complexity: EMNIST, CIFAR-100, and a combined MNIST+FashionMNIST set. Each dataset is presented sequentially to the network, with inputs encoded into spike trains and stored via our full SHN pipeline: SIM-STDP with WTA for

weight updates, TTG for memory protection, and FSW for retrieval. Performance is tested under three conditions: (i) *None*, exact recall of clean inputs; (ii) *Noise*, where 20% Gaussian noise is added; and (iii) *Masking*, where 50% of the pixels are removed. and where $\beta = 3$ is chosen to perform for our experiments.

Network sizes are set larger than the number of classes for each dataset: EMNIST (300, 1300, 2300), MNIST+FMNIST (300), and CIFAR-100 (700). Retrieval quality is measured primarily by mean squared error (MSE).

## 4.2 RETRIEVAL ANALYSIS

Table 1 summarizes average MSE across datasets, network sizes, and corruption conditions, while Figures 3 plot retrieval error over the course of sample presentation. Together these results highlight three key trends.

**Table 1:** Retrieval performance (MSE ↓) across datasets for 3 input conditions. Values are mean ± std over all samples. Lower is better

| Dataset | Size | Samples | Method | None | Noise | Mask |
|---|---|---|---|---|---|---|
| EMNIST | 300 | 3000 | Hopfield + TTG | $0.0456 \pm 0.0371$ | $0.0557 \pm 0.0500$ | $0.0649 \pm 0.0461$ |
| | | | FSW + TTG | $0.0491 \pm 0.0427$ | $0.0544 \pm 0.0483$ | $0.0529 \pm 0.0469$ |
| EMNIST | 1300 | 3000 | Hopfield + TTG | $0.1034 \pm 0.0517$ | $0.1425 \pm 0.0639$ | $0.1308 \pm 0.0464$ |
| | | | FSW + TTG | $0.1081 \pm 0.0537$ | $0.1264 \pm 0.0568$ | $0.1239 \pm 0.0555$ |
| EMNIST | 2300 | 3000 | Hopfield + TTG | $0.1422 \pm 0.0464$ | $0.1798 \pm 0.0663$ | $0.1568 \pm 0.0359$ |
| | | | FSW + TTG | $0.1496 \pm 0.0493$ | $0.1695 \pm 0.0558$ | $0.1652 \pm 0.0535$ |
| EMNIST | 300 | 3000 | Hopfield + Penalty | $0.0742 \pm 0.0327$ | $0.0839 \pm 0.0416$ | $0.0890 \pm 0.0369$ |
| | | | FSW + Penalty | $0.0773 \pm 0.0338$ | $0.0816 \pm 0.0384$ | $0.0800 \pm 0.0375$ |
| CIFAR | 700 | 2300 | Hopfield + TTG | $0.1300 \pm 0.0924$ | $0.1316 \pm 0.0926$ | $0.2183 \pm 0.0805$ |
| | | | FSW +TTG | $0.1311 \pm 0.0936$ | $0.1324 \pm 0.0937$ | $0.1303 \pm 0.0931$ |
| MNIST + FMNIST | 300 | 3000 | Hopfield + TTG | $0.1267 \pm 0.1051$ | $0.1550 \pm 0.1543$ | $0.2254 \pm 0.1621$ |
| | | | FSW + TTG | $0.1851 \pm 0.1754$ | $0.1896 \pm 0.1788$ | $0.1989 \pm 0.1873$ |

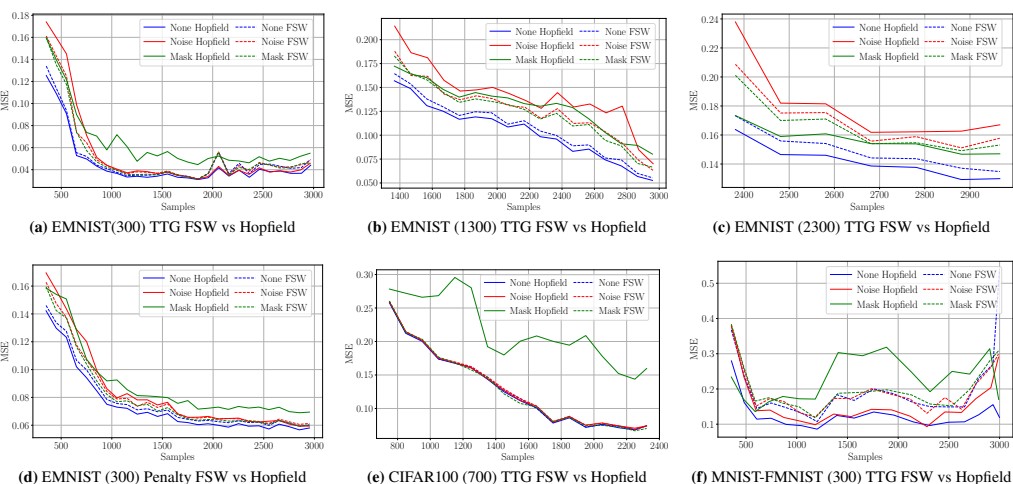

**(a)** EMNIST(300) TTG FSW vs Hopfield    **(b)** EMNIST (1300) TTG FSW vs Hopfield    **(c)** EMNIST (2300) TTG FSW vs Hopfield

**(d)** EMNIST (300) Penalty FSW vs Hopfield    **(e)** CIFAR100 (700) TTG FSW vs Hopfield    **(f)** MNIST-FMNIST (300) TTG FSW vs Hopfield

**Figure 3:** MSE performance across multiple datasets comparing FSW and Hopfield retrieval in an SHN trained with TTG: (a–c) EMNIST (300, 1300, 2300), (e) CIFAR-100 (700), and (f) MNIST+FMNIST (300). (d) shows EMNIST (300) with catastrophic forgetting controlled by the Penalty method (see Appendix A.2) instead of TTG. MSE generally decreases across datasets, except for MNIST+FMNIST where it rises after ∼1,200 samples. Hopfield recall uses $\beta = 3$ in all cases.

From Table 1, mean squared error (MSE) increases as network size grows on EMNIST: larger networks admit more patterns, but interference between them raises reconstruction loss under all conditions (None/Noise/Mask). This indicates that TTG prioritizes preserving earlier patterns, while newer inputs may be only partially accommodated once thresholds tighten—consistent with the intended catastrophic forgetting control. As seen in the same table, the penalty-based variant faithful to EWC (Appendix A.2) performs substantially worse under identical settings, since global nudging toward Fisher-weighted means blurs stored patterns, in contrast to TTG's selective gating which preserves sharp memories.

Comparing retrieval rules, FSW performs similarly to Hopfield recall on clean and noisy inputs, but shows consistent advantages under masking. This effect is strongest on CIFAR-100, where masking 50% of pixels severely degrades Hopfield recall, yet FSW maintains lower error by selecting a stable winner neuron from partial evidence.

Figures 3 provide further insight beyond the averages in Table 1. On EMNIST, MSE steadily decreases and stabilizes as more samples are stored, except in the 1300-neuron setting where error continues to decline beyond 3000 samples. In contrast, the combined MNIST+FMNIST dataset shows instability: error decreases until about 1200 samples, then rises again, with FSW particularly impacted under the *None* condition. This suggests that mixing heterogeneous datasets in one memory pool can destabilize retrieval.

Overall, the results confirm that SHN with TTG sustains recall fidelity under corruption, while FSW sharpens recovery when inputs are heavily masked. The anomalies observed in MNIST+FMNIST highlight the limits of shared storage and point to directions such as pruning or dataset separation for future work.

### 4.3 ABLATIONS STUDIES

To investigate the role of Temporal Threshold Gating (TTG), we varied the decay factor $\gamma$ in Equation 13. Table 2 reports MSE on the EMNIST dataset (size 300, 3,000 samples) under three settings: no catastrophic forgetting (no CF), $\gamma = 1$, and $\gamma = 10$.

**Table 2:** Ablation on TTG decay factor $\gamma$. Retrieval error (MSE ↓) under different corruption settings. Lower is better.

| Setting | None | Noise | Mask |
|---|---|---|---|
| Hopfield ($\gamma = 1$) | $0.0931 \pm 0.0561$ | $0.1136 \pm 0.0594$ | $0.0933 \pm 0.0447$ |
| FSW ($\gamma = 1$) | $0.1231 \pm 0.0589$ | $0.1279 \pm 0.0573$ | $0.1267 \pm 0.0577$ |
| Hopfield ($\gamma = 10$) | $0.0427 \pm 0.0348$ | $0.0527 \pm 0.0487$ | $0.0611 \pm 0.0446$ |
| FSW ($\gamma = 10$) | $0.0442 \pm 0.0373$ | $0.0496 \pm 0.0444$ | $0.0482 \pm 0.0429$ |
| Hopfield (no CF) | $0.0425 \pm 0.0344$ | $0.0523 \pm 0.0484$ | $0.0607 \pm 0.0443$ |
| FSW (no CF) | $0.0438 \pm 0.0365$ | $0.0490 \pm 0.0437$ | $0.0477 \pm 0.0423$ |

Surprisingly, $\gamma = 10$ yields the lowest error across all conditions. A larger $\gamma$ shrinks threshold $\vartheta_i$ for a particular neuron $i$ faster (refer to Equation 13 ), raising thresholds and thus *blocking most updates*. This means old memories are strongly preserved, yet recall fidelity remains high. In fact, results with $\gamma = 10$ nearly match the "no CF" case, where all updates are allowed without restriction.

By contrast, $\gamma = 1$ should in principle permit more updates, but error instead rises sharply at around 1200 samples as seen in Figure 5. This instability suggests that even a single corrupted update can poison storage, creating an attractor basin that later samples fall into. This echoes our earlier observation on MNIST+FMNIST (Figure 3): once a bad pattern is reinforced, MSE spikes rather than converging.

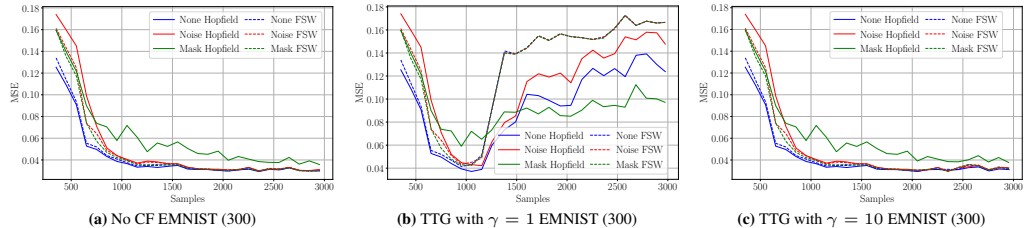

**(a)** No CF EMNIST (300)  **(b)** TTG with $\gamma = 1$ EMNIST (300)  **(c)** TTG with $\gamma = 10$ EMNIST (300)

**Figure 4:** Mean squared error (MSE) on EMNIST (size 300) under three settings: (a) no catastrophic forgetting, (b) TTG with $\gamma = 1$, and (c) TTG with $\gamma = 10$. Both (a) and (b) show MSE decay over 3,000 samples, but in (b) the MSE later increases, suggesting that corrupted data can distort performance.

Overall, this ablation confirms that TTG's effect is less about fine-tuning decay speed and more about *shielding against corrupted updates*. In practice, SHN performance depends critically on whether updates reinforce clean or faulty inputs—a property future work could mitigate with pruning or selective update strategies.

## 5 CONCLUSION

We presented a spiking Hopfield network (SHN) that integrates biologically grounded dynamics to extend the bio-realism of modern Hopfield networks (MHNs). Our design stores and retrieves patterns without gradient-based backpropagation by combining three mechanisms: sim-STDP for local weight updates, WTA for slot selection, and the First Spike Wins (FSW) rule for efficient retrieval. To mitigate catastrophic forgetting, we introduced Temporal Threshold Gating (TTG), which enforces selective updates that preserve previously stored memories.

Experiments on EMNIST, CIFAR-100, and MNIST+FMNIST confirm that SHN scales to large capacities while maintaining retrieval fidelity. FSW matches Hopfield recall under clean and noisy inputs, and consistently outperforms it under severe corruption (masking), particularly on CIFAR-100. TTG further stabilizes storage by protecting older memories, though anomalies reveal sensitivity to corrupted inputs—an effect also shared by classical Hopfield recall, underscoring the general challenge of corrupted data in associative memory.

Our results demonstrate that Hopfield-style memory can operate through spiking dynamics rather than gradient descent, offering a functional analogue to hippocampal storage and recall. This opens a pathway toward efficient, online, and neuromorphic implementations. Future extensions may incorporate pruning or hybrid strategies to handle corrupted updates and extend continual learning at scale.

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

# A APPENDIX

## A.1 SENSITIVITY TO RETRIEVAL SHARPNESS ($\beta$)

We conducted an additional sensitivity study varying the retrieval scaling parameter $\beta$. Recall that $\beta$ controls the contrast of Hopfield recall in Equation 5: smaller values produce smoother, averaged reconstructions, while larger values sharpen outputs but can amplify noise.

We compared $\beta = 0.5$, $\beta = 1$, and $\beta = 3$ (used in the main experiments). Surprisingly, smaller $\beta$ values sometimes yielded slightly lower MSE, but without a consistent advantage across conditions. In particular, $\beta = 0.5$ occasionally outperformed $\beta = 1$, yet $\beta = 3$ produced clearer and more interpretable reconstructions, which is why we standardized on $\beta = 3$ in the main experiments.

**Table 3:** Effect of retrieval scaling $\beta$ on EMNIST (300 neurons). Values are mean MSE $\pm$ std.

| Condition | $\beta = 0.5$ | $\beta = 1$ | $\beta = 3$ |
|---|---|---|---|
| None | $0.0412 \pm 0.0335$ | $0.0450 \pm 0.0342$ | $0.0456 \pm 0.0371$ |
| Noise | $0.0510 \pm 0.0457$ | $0.0531 \pm 0.0468$ | $0.0557 \pm 0.0500$ |
| Mask | $0.0615 \pm 0.0461$ | $0.0638 \pm 0.0465$ | $0.0649 \pm 0.0461$ |

These results suggest that $\beta$ mainly adjusts the tradeoff between numerical error and visual sharpness, rather than fundamentally changing retrieval dynamics. We therefore treat $\beta$ as a presentation hyperparameter rather than a core factor in evaluating our SHN.

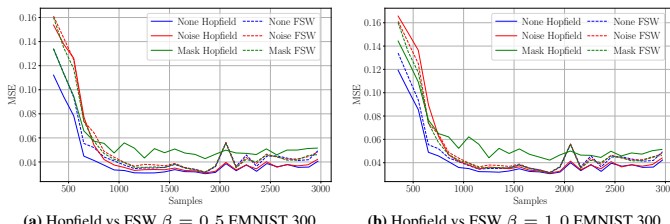

(a) Hopfield vs FSW $\beta = 0.5$ EMNIST 300     (b) Hopfield vs FSW $\beta = 1.0$ EMNIST 300

**Figure 5:** Retrieval performance of Hopfield recall and FSW on EMNIST (size 300) over 3,000 samples. Both settings show nearly identical dynamics, indicating that varying $\beta$ in this range does not significantly affect performance.

## A.2 PENALTY-BASED VARIANT OF EWC

For completeness, we also implemented a penalty-based variant more faithful to the original EWC formulation, where updates are regularized by Fisher information and snapshot means.

Let $x$ denote the current input, $\hat{W}$ the snapshot synapses, and $F$ the Fisher matrix. The update is weighted by a penalty factor $p_i$ for each neuron $i$:

$$\Delta W_i = p_i \, (x - \hat{W}_i),$$

where the penalty factor combines Fisher information and temporal decay:

$$p_i = \frac{1}{1 + \alpha \, F_i \, (W_i - \hat{W}_i)^2} \cdot \frac{1}{\text{age}_i}.$$

Update:

$$W_i \leftarrow (1 - p_i)W_i + p_i \, x.$$

Here, $\alpha$ is a scaling hyperparameter and $\text{age}_i$ increases with each update, reducing plasticity over time. This discourages overwriting weights strongly constrained by Fisher information, while still allowing limited adaptation.

The effective synaptic update becomes:

$$W_i \leftarrow (1 - p_i)W_i + p_i \, x.$$

Unlike TTG, which gates updates selectively, the penalty variant softly nudges all weights toward their means. As observed in Section. 4.2, this causes stored patterns to blur, leading to worse retrieval performance under identical conditions.

