# OpenReview forum: "Towards Biological Continual Learning with Spiking Hopfield Networks"
_ICLR.cc/2026/Conference — Submitted to ICLR 2026_

### Official Review · Reviewer_fh1P · 2025-10-23

**Soundness:** 2
**Presentation:** 1
**Contribution:** 2
**Rating:** 2
**Confidence:** 4

**Summary:**

The paper introduces a spiking neural network based variant of modern Hopfield networks, with a motivation to design a thoroughly biologically plausible memory model. This spiking Hopfield network (SHN) model is essentially enhanced with adaptive LIF neurons and local learning rules through spike-time-dependent plasticity (STDP). Additionally as a key contribution, based on a first-spike-wins retrieval rule from the SHN memory, the model also introduces a temporal threshold gating (TTG) memory protection mechanism with multiplicative threshold decays. This is presented as an adaptation of the elastic weight consolidation (EWC) method in the context of a Hopfield-STDP framework, to demonstrate how minimal catastrophic forgetting with local learning rules can be achieved.

**Strengths:**

- The paper has a clear biologically inspired motivation in its design of spiking Hopfield networks.
- Proposed TTG update rule is a novel adaptation of EWC for STDP-based weight updates. Instead of EWC-like parameter penalization, locally gating the updates based on an adaptively decaying firing threshold is an interesting approach to mitigate catastrophic forgetting.

**Weaknesses:**

- The paper essentially aims to adapt and combine several existing machine learning mechanisms in a single framework: adaptive LIF neurons, STDP, and EWC, within a modern Hopfield network. Therefore from a technical perspective, the main contributions seem to be regarding the design and adaptation choices rather than fundamentally novel methodologies (except TTG).
- Clarity and reproducibility of the paper is a big limitation factor, and it almost appears like there are large gaps in the manuscript. There is no code or experimental details are not present, although there is a lot of page-space to explain the work more in detail.

**Questions:**

- The design of the experiments are not clear. There is not much information in the paper, and it is hard to understand what Section 4.1 really implies. No details in the appendix either. This is particularly important from a reproducibility and a self-contained manuscript perspective. No details or no code is available for any of the experimental results.
- More details on the architecture, hyperparameters, parameter initializations, etc. should be provided. For instance, how are the input spike train encodings facilitated?
- There could be a bit more effort in unifying the notation of the paper across sections. In Sec 2.1, ALIF neuron model is described with $\theta(t)$ indicating adaptive firing thresholds, whereas $\theta$ is used in Sec 2.4 with EWC to denote parameters to be updated in Eq (6), and in Sec 3.4 and Algorithm 1 the adaptive threshold with multiplicative decay is now denoted with another variable $v_i$.
- The manuscript could benefit from a better coverage of existing works in this area. There is no discussion on existing studies with SNN-based memory models. Some examples:

[1] “Memory-dependent computation and learning in spiking neural networks through Hebbian plasticity”, 2023.

[2] “STDP-based Associative Memory Model on Spiking Neural Networks”, 2024.

[3] “Toward a Biologically Plausible SNN-Based Associative Memory with Context-Dependent Hebbian Connectivity." 2025.

---

> ### Author Response · Authors · 2025-11-18
> **Response to Reviewer fh1P's Weaknesses and Questions**
>
> **(1) Novelty Weakness**
>
> The novelty of our work does not lie in simply combining STDP, ALIF, or EWC-related ideas, but in how these mechanisms are reformulated and functionally integrated within a Hopfield-type associative memory. To our knowledge, this has not been explored in this setting.
>
> In our model, STDP does not train feed-forward layers; it determines which memory trace is updated at each time step within a structure consistent with the Hopfield memory space. This is what enables true online associative learning.
>
> TTG is not an adaptation of EWC. It is a distinct, fully local metaplastic threshold mechanism that stabilises stored memories without relying on global gradients.
>
> Together, these components convert a continuous Modern Hopfield formulation into a spiking, continual-learning associative memory in which storage, udpate, protection, and retrieval emerge from spike-based interaction dynamics rather than optimisation procedures.
>
> **(2) Clarity and Reproducibility Weaknesses and Question 1, 2**
>
> We appreciate the reviewer’s concern regarding experimental details. Our model does not rely on conventional parameter initialization or gradient‐based optimization.
>
> The initial synaptic state is uniform (set to –1), representing an empty memory structure that evolves directly from the presented stimuli. The number of memory neurons—which defines the associative memory capacity—is fixed and stated clearly in the experiments; it is not a tunable performance parameter as in typical SNN architectures. Multiple capacity sizes are evaluated in the experiments.
>
> For spike train duration, we use 350 time steps for MNIST-related datasets. This provides sufficient temporal resolution for spike dynamics without introducing unnecessary noise or computational cost, and will be explicitly stated in the revised manuscript.
>
> Input encoding follows a standard rate-coding scheme. SHN is intended as a general spiking associative memory framework rather than one tied to a specialised encoding strategy.
>
> These details will be clarified in the revision. Full implementation and code will be made available upon acceptance to ensure complete reproducibility. Additional clarification of the underlying MHN formulation is provided in the General Comment.
>
>
> **(3) Variable naming confusion: Question 3**
>
> We thank the reviewer for raising this point. The symbols used in Sections 2.1 and 2.4 follow the original formulations we review, and we intentionally preserve the authors’ notation. In those sections, $\theta$ denotes the adaptive firing threshold of the ALIF neuron model, and the network parameters used in EWC, respectively, as clearly specified in the manuscript and consistent with prior work.
>
> The threshold introduced in Section 3.4 serves a completely different role: it is the adaptive gating threshold $\vartheta$ used for catastrophic-forgetting mitigation in the TTG mechanism. It is not related to the neuronal firing threshold and therefore uses a distinct symbol by design.
>
> For clarity, we will make this distinction more explicit in the revision, but the current notation is already correct given the different functions of the variables.
>
>
>
> **(4) Modern Hopfield Network Discussion: Question 4**
>
> We appreciate the reviewer’s suggestions. Our work is built directly on the Modern Hopfield Network (MHN) framework, and the related work we cite focuses on the Hopfield-based associative memory paradigm that our SHN extends.
>
> The papers suggested by the reviewer are general spiking models and do not utilise Hopfield memory structures or attractor-based retrieval. As such, they address a different problem setting and are not directly related to the design or objective of SHN.
>
> For completeness, we will briefly mention these works in the revised manuscript, while keeping the main related-work discussion centered on Hopfield and MHN literature, which forms the correct theoretical foundation for SHN.

---

### Official Review · Reviewer_9g8H · 2025-10-27

**Soundness:** 2
**Presentation:** 1
**Contribution:** 2
**Rating:** 4
**Confidence:** 2

**Summary:**

They authors derive both learning and update rules for a spiking Hopfield network that allows ongoing learning without catastrophic forgetting, and is relatively insensitive to noise.

**Strengths:**

The authors address an important problem, at least from a neuroscience perspective: spiking Hopfield networks. The experimental results were very good, especially since they were learning in an online setting -- something that is difficult for associative memory networks.

The explanation of the individual parts was, by and large, understandable.

**Weaknesses:**

I might be missing something (not unusual), but I could not figure out what they actually did. The problems start with Eq 1,

tau_m du/dt = -u + RI.

However, we were never told what the current, I was. Therefore, I could not figure out what the architecture was, or how the data, x, was presented to the network. Each of the subsections by themselves made sense; I just couldn't figure out how they fit together.

**Questions:**

Please tell me what the architecture was. I can't guarantee it, but I'm almost positive I'll raise my score once I understand the architecture. Assuming I don't find something else wrong, which I doubt will happen.

Also, I wasn't clear on the training: how many times was each input presented? And was the MSE calculated on a training set?

Typo, line 359 (I think)o :  Figure 5 should be  Figure 4.

---

> ### Author Response · Authors · 2025-11-18
> **Response to 9g8H's Architectural concerns and Question**
>
> We thank the reviewer for the feedback.
>
> We clarify that the variable \(I\) in Eq.1 represents the input derived from the stimulus at each time step. It is not an architectural module but the basic drive to the membrane potential \(u\), which determines when a neuron fires. This is the standard formulation defining spiking neuronal behaviour in SNNs.
>
> Our SHN, however, should not be mistaken for a conventional SNN or autoencoder. It functions as an associative memory system in which spiking dynamics govern both storage and retrieval. The overall structure follows the classical associative‐memory form: the input pattern projects onto a finite set of memory neurons representing stored attractors within a synaptic weight matrix. This architecture is clearly illustrated in Figure 2b. See the General Comment for details on the biologically inspired design and implementation.
>
> The reviewer also raised uncertainty about the training procedure. Our SHN operates in an online learning regime rather than a batch or supervised setting. Each input is presented once and processed individually, and the network immediately updates the corresponding memory synapses through our local STDP‐like algorithm without backpropagation. Evaluation occurs concurrently by measuring convergence error after each presentation, consistent with standard online associative‐memory protocols.

---

> > ### Comment · Reviewer_9g8H · 2025-11-18
> > **Reply from 9g8H**
> >
> > Sorry -- your answer barely helped, in the sense that I still don't know what equations you actually implemented. Let me be more clear about my confusion.
> >
> > Eq. 1 introduces a membrane potential, u. (Which, presumably, is really u_i for neuron i?)
> >
> > Eq. 2 defines the threshold, theta(t). (Which, presumably, should also have a subscript i?)
> >
> > Eq. 3 introduces a learning rule for updating the weights, w_ij. But I don't know what role w_ij plays in the circuit.
> >
> > Eq. 4 introduces an energy involving X and xi.
> >
> > Eq. 6 introduces an "augmented loss" involving theta_i. Which I don't think are thresholds, but I'm not sure.
> >
> > My problem is that I don't know how these variables fit together. Could you please provide the full set of equations? I'm guessing Fig. 1a is supposed to tell me everything, but I've never been good at turning figures into equations.

---

> > > ### Author Response · Authors · 2025-11-27
> > > **Comment on the Follow-Up Response**
> > >
> > > We thank the reviewer for the follow-up.
> > >
> > > **Interpretation of the Preliminaries**
> > >
> > > As noted earlier, the reviewer’s difficulty appears to come from interpreting the preliminaries (Sec. 2) as if all equations (1)–(6) were intended to form a single unified ANN-style architecture. They are not. Each equation originates from a different foundational component, and their meanings are fully defined in the cited literature. They are included to provide theoretical context, not to describe a single combined computational circuit.
> > >
> > > **Regarding the equations**
> > >
> > > Questions about “how they fit together’’ were already addressed in the general comment, which summarises the actual SHN architecture at the correct level of abstraction.
> > >
> > > *  Eqs. (1)–(2) describe standard single-neuron spiking dynamics, and Eq. (3) demonstrates a local plasticity rule within an SNN. Reading them through ANN logic and indexing conventions naturally leads to the misconceptions reflected in the reviewer’s follow-up.
> > >
> > > * Eqs. (4) and (6) (MHN retrieval and EWC) are background mechanisms motivating our design, not components executed jointly in a single forward pass.
> > >
> > > The complete set of equations instantiated in SHN already appears in the manuscript. We will add a brief note in the revision indicating which preliminaries are used directly. Beyond this, further explanation of ALIF, MHN, STDP, and EWC belongs to established theory already covered in the referenced works.

---

> > > > ### Comment · Reviewer_9g8H · 2025-11-28
> > > > **Reply 2 from 9g8H**
> > > >
> > > > I'm perfectly happy to have the area chair over-rule me, but I'm not going to crawl through the literature to figure out what you actually did. Especially when it would probably take you about 15 minutes to write down the actual equations describing the network.
> > > >
> > > > Sorry -- possibly a culture clash here.

---

### Official Review · Reviewer_mcpL · 2025-10-31

**Soundness:** 2
**Presentation:** 2
**Contribution:** 2
**Rating:** 2
**Confidence:** 4

**Summary:**

This paper introduces a Spiking Hopfield Network (SHN) that implements a biologically plausible version of  Modern Hopfield Networks by adding spiking dynamics and neural learning rules. To do so, they add the following to their network, which uses adaptive leaky integrate-and-fire neurons: (1) a simplified STDP-like learning rule, (2) a first-spike-wins retrieval mechanism for memory recall instead of softmax retrieval (3) temporal threshold gating for synaptic updates to prevent catastrophic forgetting. They compare SHN + first-spike-wins with SHN + Hopfield retrieval and show that the former does comparably on EMNIST, CIFAR-100, and MNIST+FashionMNIST tasks with sequential learning.

**Strengths:**

- The paper uses a variety of datasets (EMNIST, CIFAR, MNIST/FMNIST) , going beyond random binary patterns.
- Given that modern Hopfield networks are often connected to long term memory in brains, the problem of understanding what it takes to make them more biologically plausible is important to researchers studying memory in neuroscience.

**Weaknesses:**

- The use of winner-take-all dynamics for storage, but then first-spike-wins for retrieval feels inconsistent. Also first-spike-wins seems unusual as a biological mechanism. It's unclear to me if there's biological evidence for it.
- The only comparison model shown is SHN + first-spike-wins vs SHN + Hopfield retrieval. Feels like there should be a comparison to the standard, non-spiking MHN. Also, because the comparisons are only done between two versions of SHN and there's no visual examples shown of the retrieved memory, it's hard to interpret the MSE reported.
- I would have liked more discussion about what neuroscience insights can be gained by the design of a spiking MHN. The lack of the discussion, combined with some of the arbitrary choices of design and the difficulty of interpreting the performance of the model (the two points mentioned above), makes it hard to evaluate the contribution made by the paper.

**Questions:**

- How does the model perform compared to MHNs?
- Can you discuss/justify more the choice of using two mechanisms for slot selection and retrieval (WTA and FSW, respectively)?

---

> ### Author Response · Authors · 2025-11-18
> **Response to mcpL’s Weaknesses and Questions**
>
> **(1) WTA vs. FSW and Question 2**
>
> We would like to clarify that WTA for storage and FSW for retrieval serve distinct and complementary functions within our SHN.
>
> During storage, WTA operates under our STDP‐based update rule pinpoints a memory unit—among a finite set of memory slots— with the highest similarity to the current input temporally at each time step. Only this winning memory trace is updated, while all others remain unchanged, preserving existing patterns and preventing interference.
>
> During retrieval, instead of integrating spike responses across all time steps, FSW relies on the earliest spike as the decision signal: the first neuron to fire identifies the most similar stored pattern. This mechanism, supported by biological evidence of latency coding (Zohar et al.; Chase et al.), shortens retrieval time and improves efficiency while maintaining recall fidelity—a clear advantage rather than a weakness.
>
> Therefore, WTA and FSW are not alternative selection rules but phase‐specific processes: WTA refines storage and updating, while FSW enables efficient and biologically plausible retrieval.
>
>
>
> **(2) MHN retrieval evaluation and Question 1**
>
> We clarify that all experiments were already compared against the standard Modern Hopfield Network (MHN) retrieval rule, which uses a softmax and vector‐based similarity for pattern recall.
>
> Our SHN replaces this mechanism with a biologically inspired spiking‐dynamic retrieval process—a novel approach that achieves comparable or better convergence stability, particularly under 50\% masking conditions.
>
> This confusion may stem from interpreting the Hopfield baseline as an autoencoder‐like model, whereas it is in fact the standard MHN retrieval algorithm evaluated under a continual learning setting.
>
> These mechanisms are not ad-hoc or interchangeable components; each was purposefully developed to embed spiking dynamics within the associative memory framework of a Hopfield network. The design choices follow directly from the goal of achieving biologically grounded storage and retrieval, as detailed in the general comment.
>
>
> **(3) Biological Design and Motivation**
>
> Please refer to our general comment, where we clarified the biological motivation, implementation, and correspondence of STDP, TTG, and FSW to hippocampal-inspired memory processes and plasticity mechanisms within the MHN framework.

---

### Author Response · Authors · 2025-11-18
**General Comment to Reviewers and Area Chair**

We thank the reviewers and the area chair for their time. Several comments indicate that key aspects of Modern Hopfield Networks (MHNs) and the intended scope of our Spiking Hopfield Network (SHN) may have been interpreted through the lens of feed-forward or autoencoder-style models. Notably, none of the critiques questioned the core novelty of our contribution. To avoid any ambiguity, we restate the central distinctions below:

* An MHN is an associative memory system, not a function approximator. Information is stored in the synaptic weights and retrieved through attractor dynamics (Hopfield; Krotov et al.). While prior work such as Ramsauer and Hochreiter treates MHN as a geometric vector-space representation, our SHN reformulates the MHN as a spiking, hippocampal-inspired architecture where recall emerges from biological interaction dynamics.
* Our custom STDP rule is the backbone of the model: it determines memory similarity, recall, and synaptic updates in a biologically consistent manner rather than through vector operations. It is not a mechanism for adjusting layer weights to map inputs to outputs, as in autoencoders.
* The temporal threshold gating (TTG) mechanism provides local, EWC-like stability control that preserves stored memories. This component is essential and new for online associative learning in SNNs without global gradients.
* Experiments follow standard protocols for online associative-memory Hopfield networks and demonstrate convergence and stability: our SHN reliably reaches valid attractor states as memory size increases.

These distinctions will be stated explicitly in the revised version.

---

### Meta-Review · Area_Chair_ogTG · 2026-01-03

**Summary:**

The paper proposes a Spiking Hopfield Network (SHN) for online associative learning. The authors' rebuttal clarified that the model operates within the Modern Hopfield Network (MHN) framework and should not be evaluated as a standard feed-forward SNN or Autoencoder.  However, The fact that multiple reviewers fundamentally misinterpreted the model’s category and scope indicates a significant issue with the manuscript's clarity and presentation. Therefore, the paper remain requires significant rewriting to improve its positioning and readability.

**Reviewer Concerns:**

The main outstanding issue is the lack of clarity. for exanple, the distinction between the proposed SHN and conventional SNNs/Autoencoders was not made sufficiently clear.

**Reviewer Scores:**

I believe the reviewers' scores would likely remain unchanged

---

### Decision · Program_Chairs · 2026-01-26

Reject